# Sexual dichromatism in the neotropical genus *Mannophryne* (Anura: Aromobatidae)

**Mark S. Greener**[1]*, **Emily Hutton**[2], **Christopher J. Pollock**[3], **Annabeth Wilson**[2], **Chun Yin Lam**[2], **Mohsen Nokhbatolfoghahai**[2], **Michael J. Jowers**[4], **J. Roger Downie**[2]

**1** Department of Pathology, Bacteriology and Avian Diseases, Faculty of Veterinary Medicine, Ghent University, Merelbeke, Belgium, **2** School of Life Sciences, University of Glasgow, Glasgow, United Kingdom, **3** School of Biology, Faculty of Biological sciences, University of Leeds, Leeds, United Kingdom, **4** CIBIO/InBIO (Centro de Investigacao em Biodiversidade e Recursos Genticos), Universide do Porto, Vairao, Portugal

* mark.s.greener@gmail.com

**Data Availability Statement:** All relevant data are within the paper and its Supporting Information files.

## Abstract

Recent reviews on sexual dichromatism in frogs included *Mannophryne trinitatis* as the only example they could find of dynamic dichromatism (males turn black when calling) within the family Aromobatidae and found no example of ontogenetic dichromatism in this group. We demonstrate ontogenetic dichromatism in *M. trinitatis* by rearing post-metamorphic froglets to near maturity: the throats of all individuals started as grey coloured; at around seven weeks, the throat became pale yellow in some, and more strongly yellow as development proceeded; the throats of adults are grey in males and variably bright yellow in females, backed by a dark collar. We demonstrated the degree of throat colour variability by analysing a large sample of females. The red: green (R:G) ratio ranged from ~1.1 to 1.4, reflecting variation from yellow to yellow/orange, and there was also variation in the tone and width of the dark collar, and in the extent to which the yellow colouration occurred posterior to the collar. Female *M. trinitatis* are known to be territorial in behaviour. We show a positive relationship between throat colour (R:G ratio) and escape performance, as a proxy for quality. Our field observations on Tobago's *M. olmonae* showed variability in female throat colour and confirmed that males in this species also turn black when calling. Our literature review of the 20 *Mannophryne* species so far named showed that all females have yellow throats with dark collars, and that male colour change to black when calling has been reported in eight species; in the remaining 12 species, descriptions of males calling are usually lacking so far. We predict that both dynamic and ontogenetic sexual dichromatism are universal in this genus and provide discussion of the ecological role of dichromatism in this genus of predominantly diurnal, non-toxic frogs, with strong paternal care of offspring.

## Introduction

Most anuran amphibians are active mainly at night and intra-specific communication is mainly mediated by acoustic signals [1]. However, as more complexity in anuran behaviour is

**Funding:** Funder information: Dennis Curry Charitable Trust University of Glasgow Chancellor's Fund Percy Sladen Memorial Fund Glasgow Natural History Society Gilchrist Educational Fund Due to the nature of the University of Glasgow Exploration Society Expeditions, funding is not awarded to people or for the purpose of conducting research for publication. It is awarded to the Trinidad Expedition to allow undergraduate students to conduct and experience science first-hand. Therefore, the authors were not specifically funded. I have detailed in the information above, funders of the Trinidad Expedition that aided in the facilitation of the expeditions. I am unsure if this would class these studies as unfunded, as they were not the purpose of the expeditions, not specifically funded and the expedition would have continued regardless of our studies. "The funders had no role in study design, data collection and analysis, decision to publish, or preparation of the manuscript." "The author(s) received no specific funding for this work.

**Competing interests:** The authors have declared that no competing interests exist.

found, a wide diversity of visual signalling (movements, colours, patterns, shapes) both in diurnal and in nocturnal species is becoming established. For example, Rojas [2] reviewed the roles of colours and patterns; Hödl and Amezquita [3] reviewed and classified the variety of visual signals, and Starnberger *et al*. [4] discussed the multimodal roles of the vocal sac in signalling: not only auditory, but also visual and chemical in some cases. One category of visual signals involves sexual dichromatism, reviewed by Bell and Zamudio [5]. They distinguished two types. First, dynamic dichromatism, restricted to males, where the male develops a temporary colour signal related to courtship and breeding. The review identified 31 species in nine families where this occurred. Second, ontogenetic dichromatism, where either males or females develop a permanent colour difference as they mature. The review found this reported from 92 species in 18 families. Bell *et al*. [6] extended the dataset for dynamic dichromatism to 178 species in 15 families and subfamilies. Bell and Zamudio [5] found no species reported as having both sexual and ontogenetic dichromatism. However, a recent paper by Engelbrecht-Wiggans and Tumulty [7] has documented both in the aromobatid frog *Anomaloglossus beebei*.

Dynamic sexual dichromatism is likely to be an aspect of sexual selection, where the male's temporary colour in some way attracts females. Most frogs attract mates through acoustic signals and breed at night, when a colour signal would appear not to be useful. However, in some diurnal species there is evidence that colour can be used by females to assess male quality in nocturnal breeding aggregations where discrimination of acoustic signals is difficult [8]. This may not always be the case, as in *Rana arvalis* that undergo a temporary colour change to blue during explosive breeding events for several days of the year, it was found that colouration was used to used for sex recognition by males [9].

Bell and Zamudio [5] included two classes of ontogenetic dichromatism. In the first, where males develop more conspicuous colouration than females, sexual selection is likely to be the main driver, with an expectation that the permanent colour should not be at significant cost. An example is the poison frog *Oophaga pumilio* where the male's bright colour attracts females but also acts as an aposematic signal, deterring predators [10]. In the second class, females develop brighter colours than the males: this may be explicable through sexual selection, but an alternative may be sexual niche partitioning, where the two sexes occupy different niches, and colour contributes in some way to successful occupation.

In this paper, we report on the occurrence of both dynamic and ontogenetic sexual dichromatism in frogs of the neotropical genus *Mannophryne*. Frost [11] lists 38 species in the subfamily Aromobatinae of which 20 belong to the genus *Mannophryne*. *Mannophryne* are ground-living frogs inhabiting the environs of mountain streams in Venezuela and the Caribbean islands of Trinidad and Tobago (West Indies). They are cryptically coloured, with dorsal sides mostly mottled grey and brown, and they lack the poison gland protection found in dendrobatids, to which they are closely related [12,13]. Most accounts find *Mannophryne* to be day active frogs, but there are occasional reports of them remaining active after dusk [14,15, this paper].

We focus on sexual dichromatism in two species, *M. trinitatis* and *M. olmonae*, but also review literature reports from other *Mannophryne* species. Kenny [16] noticed that when male *M. trinitatis* are calling, their colour changes from the normal cryptic mottled brown/grey to jet black. Wells [17] reported that the colour change is very rapid, occurring over 1–10 minutes, both at the start of calling and at the end of an episode of calling when the colour reverts to normal. Bell and Zamudio [5] included this as their only example of dynamic sexual dichromatism in the family Dendrobatidae (we follow Frost [11] in placing *Mannophryne* in the family Aromobatidae). Wells [17] noticed another form of sexual dichromatism in *M. trinitatis*. Females aggressively defend territories and display their pulsating bright yellow throats when they do so. The yellow patch is posteriorly bounded by a narrow dark collar (a pigmented

band of variable tone and width, extending across the ventral surface at the level of the fore-limbs). Males possess the collar, but the throat is grey and Wells comments that it is not pulsated during aggressive encounters between males. Bright yellows are often based on carotenoid pigments, regarded as costly to synthesise, and in many taxa, including some anurans, they have been associated with signals of quality (reviewed by Olson & Owens [18]; tree frog example: Richardson *et al.* [19]).

In this paper, we assess the variability of the yellow pigmentation in a large sample of *M. trinitatis* females, and report on an experiment where we used escape responses as a proxy for quality. We also document the development of the yellow throat colour as metamorphs grow towards maturity. Finally, we provide field observations on the behaviour of both *M. trinitatis* and *M. olmonae*, and review what has been reported on sexual dichromatism in other *Mannophryne* species.

## Materials and methods

### Colour variability in female *Mannophryne trinitatis*

A large sample (n = 500) of adult female *M. trinitatis* was collected from different localities (Fig 1, S1 Table) in Trinidad's Northern Range during June to August 2015 and 2016. Females, recognised by having yellow throats, were caught either by hand or with the aid of small hand-nets and placed for transport into small polyethylene bags containing a little damp forest leaf litter. They were kept overnight in a holding tank furnished with damp leaf litter, at the University of the West Indies, St. Augustine. The frogs were photographed and measured (snout-vent length [SVL] to 0.1 mm using dial callipers) on the day after collection, and then returned to their collection sites within ~20m of capture location. Despite an earlier report of the presence of the potentially deadly fungal pathogen *Batrachochytrium dendrobatidis* (*Bd*) in the *M. trinitatis* population, Greener *et al.* [20] found the infection to be absent. Capturing and returning the frogs should therefore not risk spreading infection.

To photograph the throat region, each frog was carefully held upside down and the ventral side photographed on a white background beside an X-Rite Colorchecker passport colour rendition chart (X-Rite Inc., Michigan, USA). All photographs were taken using a Sony ILCE-6000 with an attached Sony E-Mount SEL 55-200mm lens and lens hood (2015) or SEL 18-55mm lens (2016). The camera was set with F8.0 aperture, and shutter speed adjusted to give an exposure of 0.0. Photographs were taken in a white light box studio. Constant light was provided by three LED lights positioned on the right, left and above, along with an overhead strip light. All batches (frogs caught that day) of photographs were taken within one hour of each other. As in Stevens *et al.* [21], RAW format was chosen, as opposed to JPEG–as used by Bergman and Beehner [22]–as this prevents loss of information due to compression, allows for later adjustment, and due to current storage capabilities, file size was not an issue. The photograph from the first frog in each batch of frogs was used to create a colour profile using the software accompanying the X-Rite Colorchecker. A profile allows for colour correction given the same conditions. This profile was then applied to all photographs within the batch in Adobe Photoshop CC 2015, and white balance corrected for each photograph. The area of interest (yellow throat region) was selected, and average blurred before red (R) and green (G) values were recorded. These values were then used to generate R:G ratios. The presence of yellow colouration under the collar was also qualitatively assessed as Yes or No. We modelled the relationship between R:G ratio and SVL, site and presence of colouration under the collar using generalized linear mixed models, including year as a random effect, with statistical significance evaluated by Wald chi-squared testc. All analysis was conducted in RStudio 1.2.5033 environment R3.6.2 [23].

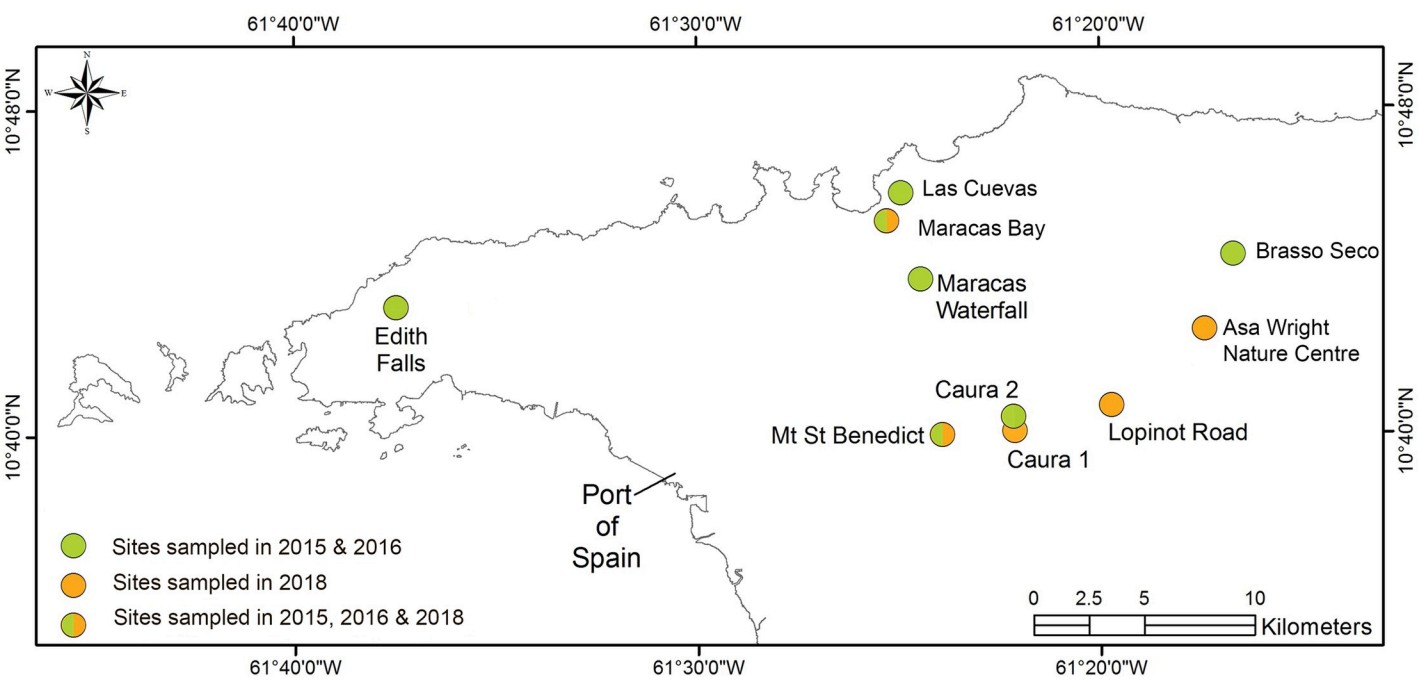

**Fig 1. Map of sites used in field studies.** *M. trinitatis* sampling sites, as detailed in S1 Table. Adapted from Greener *et al.* [20].

## Colour development in juvenile *Mannophryne trinitatis*

A sample of about 60 *M. trinitatis* tadpoles was collected in early July 2016 by hand netting from a pool in a stream beside the Arima-Blanchisseuse Road in Trinidad's Northern Range mountains. This was a sample of a larger number of tadpoles within the pool. The large number of tadpoles implies that many clutches had been deposited and therefore our sample had offspring from multiple parents. The tadpoles were transferred to the University of Glasgow, Scotland in two-litre polyethylene containers with the tadpoles resting on damp cotton cloth. Downie and Smith [24] showed that these tadpoles survive well under such conditions. In Glasgow, the tadpoles were grown at an initial density of 30 individuals per tank in plastic aquaria 32x18x18 cm in dechlorinated tap water at a depth of 10 cm and with a constant air supply delivered through a submerged air-stone at one end of the tank. The room where the aquaria were located had a 12:12h light/dark cycle and an air temperature of 23–24˚C. The tadpoles were fed daily with aquarium fish food flakes (New Era brand), and the water was changed weekly to avoid the build-up of waste. Each day, the aquaria were checked for tadpoles showing forelimb emergence, the sign of metamorphosis beginning. The first metamorph was found on 8th July and the last on 16th September. Each metamorph was caught by hand-net and transferred to an individual translucent 22x15x8 cm polyethylene container with an opaque lid. Each container was provided with a 'shelf' of washed gravel at one end, about 18 cm wide and 1–2 cm deep, and dechlorinated tap water to a depth of about 0.5 cm. Downie *et al.* [25] have shown that *M. trinitatis* take 6–7 days to complete metamorphosis and that they hide in gravel during that time.

After completion of metamorphosis, each froglet was provided twice a week with live *Drosophila melanogaster* as food, supplied in 5 cm long tubes, which the frogs could enter to forage. Froglets were measured (SVL, using callipers accurate to 0.1 mm) and their throat patterns noted and/or photographed, starting about two weeks after forelimb emergence and continuing at approximately every four weeks for 16 weeks. For this purpose, froglets were

captured by hand, transferred to a transparent 9 cm diameter petri dish, and measured from below. Once the 16-week growth period was complete, each frog was transferred to a communal tank for rearing to adulthood.

## Escape responses in *Mannophryne trinitatis* in relation to female throat colour

Adult female (n = 81) *M. trinitatis* were captured with the aid of small hand-nets from five sites across Trinidad's Northern Range (Fig 1, S1 Table), during mornings over six weeks, June to August 2018. Sites were visited in rotation, with 5–6 frogs captured at each visit. Frogs were transported to our laboratory at the William Beebe Tropical Research Station (Simla) and housed individually in plastic aquaria furnished with a thick layer of damp forest leaf litter. Frogs were measured (SVL to 0.1 mm with dial callipers; weight to 0.01g using an electronic balance) and any with SVL <16mm were classed as juveniles and excluded from further study. The day after collection, frogs were photographed in the morning around 09.00h and repeated at night around 21.00h: three pictures of the throat were taken using a Canon PowerShot s110 digital camera, in a white light box studio with two LED lamps providing constant light from either side of the light box. The camera was set with F3.2 aperture, and shutter speed adjusted to give an exposure of 0.0, allowing for comparison with throat colour variation measured earlier. Dorsal sides of each frog were also photographed to identify frogs, so as to ensure that no frog was used more than once, following subsequent collections (dorsal patterns are individually variable). Throat colour, as R:G ratio, was measured as described earlier.

Escape responses were assessed two mornings after collection (09.00–12.00h) in a specially constructed outdoor arena set in a shaded area: this had wooden sides about 0.8m high (to prevent frogs escaping) and enclosed an area of short grass 1.5x1.5m. Each frog was put into a 9cm diameter plastic petri dish and placed at the centre of the arena; it was left there for 30 seconds with the lid off to acclimatise. If the frog jumped before 30 seconds were over, it was recaptured and left a further 30 seconds. The frog was then stimulated by a light tap to the rear using a metre stick. Each frog's responses, three times for each frog, were recorded for 20 seconds using a GoPro HERO6 video camera set above the arena. After each response, each frog was given at least 30 minutes to recover before being stimulated to jump again. Air temperature ($^0$C) and relative humidity (%) within the arena were recorded using an ETI pocket thermo-hygrometer at the same times as each set of responses.

Image J (v1.52a) was used to measure the distance of each jump. From the recordings, we calculated maximum and minimum distances of each jump made, mean distance per jump and the total distance travelled in each trial (i.e. different measures of escape performance). Using RStudio 1.2.5033 environment R3.6.2 [23], general linear models were used to test relationships between escape performance and size (SVL), site, humidity, temperature and throat colour (as R:G ratio). The Benjamani–Hochberg procedure was used to correct for multiple testing. We expected escape performance (for example total distance jumped, or initial jump length) to be positively related both to size and throat colour, but to be independent of collection site, humidity and temperature.

## New field observations on *Mannophryne trinitatis*

While assessing the population status of *M. trinitatis* for Greener *et al.* [20], we made occasional interesting observations relevant to colour and behaviour, both during the day and at dusk/night, which we present here.

### Sexual dichromatism in *Mannophryne olmonae*

Field observations on *M. olmonae* were made at several small un-named streams in northeast Tobago in June to August 2014 and 2015. In 2015, female *M. olmonae* were captured using hand-nets and transferred in individual containers to accommodation in Charlotteville. Here, they were photographed alongside an X-rite Colour Checker rendition chart, under identical lighting conditions, using a Canon EOS Rebel T3i. Throat colour, as R:G ratio, was measured as described earlier and frogs were returned to their collection sites. As in the case of *M. trinitatis*, Thomson *et al.* [26] have shown *Bd* to be absent from the *M. olmonae* population.

### Comparison of sexual dichromatism across the genus *Mannophryne*

We performed a literature review on male and female colours in life in all *Mannophryne* species so far described. The black colour in calling males can only be seen when observing males calling in the field; the female yellow throat colour fades in preservative. In some cases, colours in life are not presented in the original species descriptions, but we were often able to find later accounts of colours in life.

### Ethics statement

A field research permit (Special Game License) was provided by Government's Wildlife Section for the years of this study (2015, 2016 and 2018) for the months of June–August, no numbers are allocated. This permit allowed for the capture and use of animals in these studies for frogs of all species up to 500 adult individuals per year. Export of tadpoles was granted by Special Export License 001192 (29/6/16). The Special Export License we received from the Trinidad Wildlife Division allowed for capture of animals and the collection of 100 animals, with no distinguishing between adults or tadpoles, it also did not require them to be returned to the wild. No ethical approval was required as studies were non-invasive and very lows stress levels and considered under the threshold of the Animals (Scientific Procedures) Act. All adult frogs (apart from a small number that died in captivity) were returned within a couple of days to their original location. Captured tadpoles were exported live to the UK, where they were reared to metamorphosis until adult. After this, they were donated to an experienced tropical frog breeder. No aspects of our study required anaesthesia, euthanasia or any kind of animal sacrifice.

## Results

### Throat colour variability in female *Mannophryne trinitatis*

Larger females were found to be more likely to have a higher R:G ratio on their throat patch ($p$ <0.001) (Fig 2). Two of our sites, Edith Falls and Las Cuevas, were found to have a significant positive effect on R:G ratio ($p$ <0.001). Females with high R:G ratios were found to be more likely to have colour posterior to the collar ($p$ 0.037). However, SVL and its interaction with R:G ratio were not significant predictors of this. We qualitatively observed that the colouration of the throat area differed between individuals not only in R:G ratio but also in the relative size of the throat patch, the intensity and width of the dark collar, and the extent to which the yellow colour extended posterior to the collar (Fig 3).

### Colour development in juvenile *Mannophryne trinitatis*

Forty-four of the tadpoles reached metamorphosis. Of these, 25 developed long enough for their sex to be distinguished by throat colour differences, 20 as females and five as males. This sex ratio is significantly biased towards females (chi squared = 15.8; $p$< 0.001). Table 1 shows

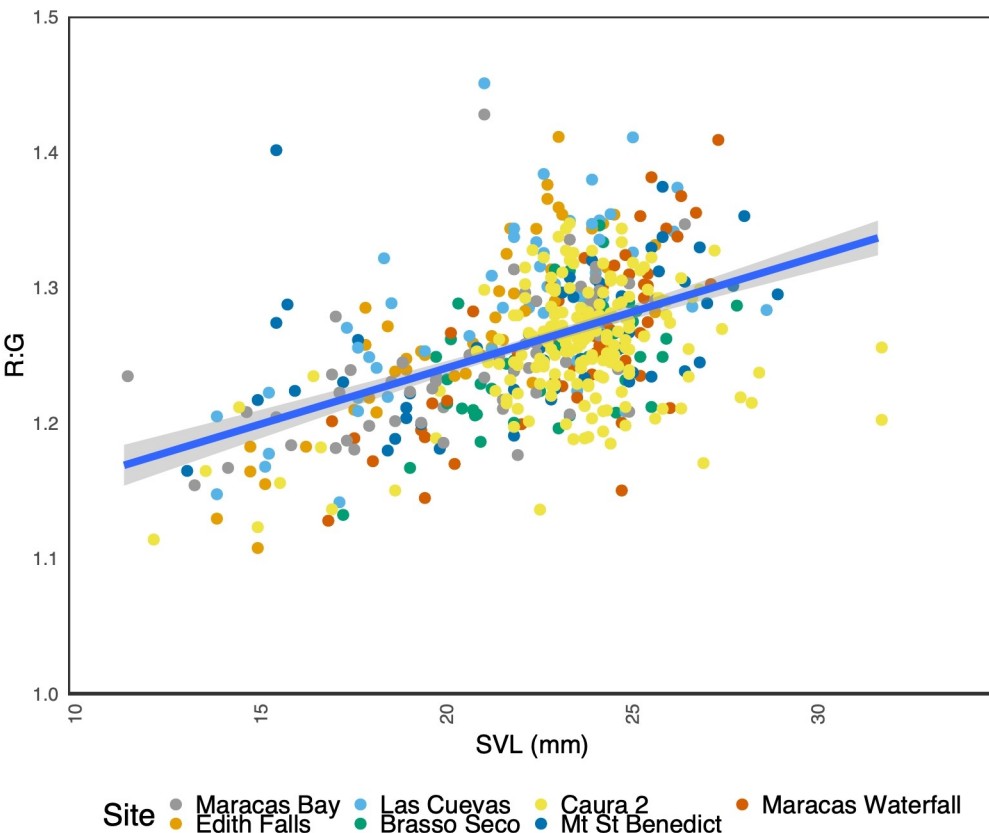

**Fig 2. The relationship between throat patch R:G ratio and SVL (mm) in *M. trinitatis* for all sites.** Line indicates linear regression of all sites combined. The shaded area indicates the 95% confidence interval.

size and colour development data for all froglets recorded beyond 90 days post metamorphosis (omitting two females that escaped at around 80 days). Dark collars developed in both males

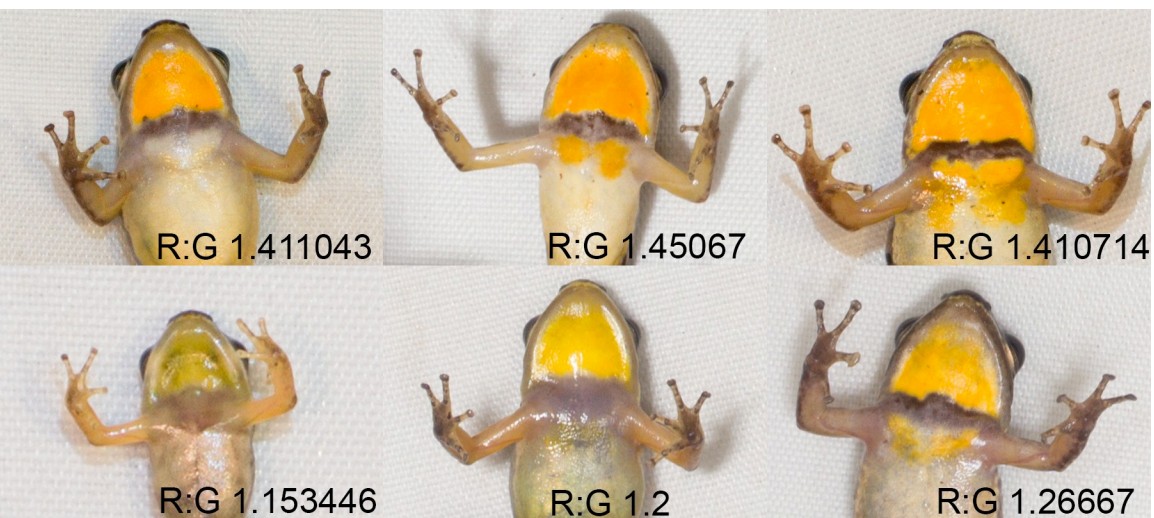

**Fig 3. Variation in the throat colouration of female *M. trinitatis*.** Variability in throat patch relative size and shape, collar width and intensity, and extension of the colour posterior to the collar. R:G denotes the Red:Green ratio as extracted from photographs.

**Table 1. The appearance of coloured throats in post-metamorphic *M. trinitatis* reared in captivity.**

| Sex | Days followed post metamorphosis (mean +/- SD | Final SVL (mm:mean +/- SD) | Yellow colour first seen | |
| --- | --- | --- | --- | --- |
| | | | Days | SVL |
| Females (n = 18) | 110.2 +/- 12.0 | 16.4 +/- 1.0 | 52.2 +/- 11.1 | 15.5 +/- 1.0 |
| Males (n = 5) | 132.4 +/- 8.2 | 17.3 +/- 0.9 | NA | NA |

and females. Although these were variable in width and shade, we saw no consistent difference between males and females. Throat colour started and remained grey in five individuals throughout the observation period and these were classified as males (we did not check sex by examining gonads). In individuals developing as females, throat colour started as grey, became pale yellow at around seven weeks post metamorphosis, and either remained pale or became more brightly yellow around nine weeks (Fig 4). Since we did not assess throat colour weekly (too frequent disturbance could be stressful, and risked escapes), we cannot tell precisely when the yellow colour first became apparent. However, throats were pale grey in all individuals at the first set of observations (2–3 weeks after metamorphosis began) and remained grey at 5–6 weeks in individuals that developed as females. These results indicate that throat colour is ontogenetically sexually dichromatic, developing in the juvenile phase, well before female maturation (mature females are around 20mm SVL; the yellow throat was distinguishable at around 15.5 mm).

## Escape responses in *Mannophryne trinitatis* in relation to female throat colour

Responses to stimulation were quite varied. Some frogs made only a few jumps before stopping on the grass; others jumped to the edge of the arena and climbed some way up the wall (*Mannophryne* have adhesive toe pads); the direction of jumping was also variable but most tended to maintain more or less the same direction once they set off.

Comparison of morning and night photographs of frog throats indicated that the colour was stable, with no diurnal variation. Table 2 shows female frog sizes, colour variability and escape performance. There were no significant differences between collection sites and escape performance ($p > 0.05$ for all measures). Air temperature during the trials had a range of only 1.8˚C (26.9–28.7˚C), but humidity varied more widely (53–86%). There were no significant relationships between temperature or humidity and escape performance (all $p > 0.05$). Also, there was no significant relationship between collection site and throat colour (R:G ratio: $p > 0.05$). However, there were strong and significant positive relationships between R:G ratio and frog size (weight: $F = 18.42$, $p < 0.001$; SVL: $F = 19.07$, $p < 0.001$). Weight and SVL were also strongly correlated with one another (not shown).

We found a positive significant relationship between R:G ratio (throat colour) and the maximum distance travelled in a single jump ($F = 11.55$, $p < 0.01$). One observation of throat colour (R:G = 1.61) had a noticeably larger R:G ratio than the rest (Fig 5), though this difference was not alarming as the relationship between maximum distance travelled and throat patch colour was still significant whether this datapoint was included or not for statistical analysis ($p < 0.01$ in both cases). There were no significant relationships between R:G ratio and the other measures of escape performance: minimum distance in a single jump; mean distance per jump; initial jump distance; total distance travelled ($p > 0.05$ in all cases).

Interestingly, we found no significant relationships between SVL and any of the escape performance measures ($p > 0.05$ in all cases). S2 Table summarises the statistical results on escape performance.

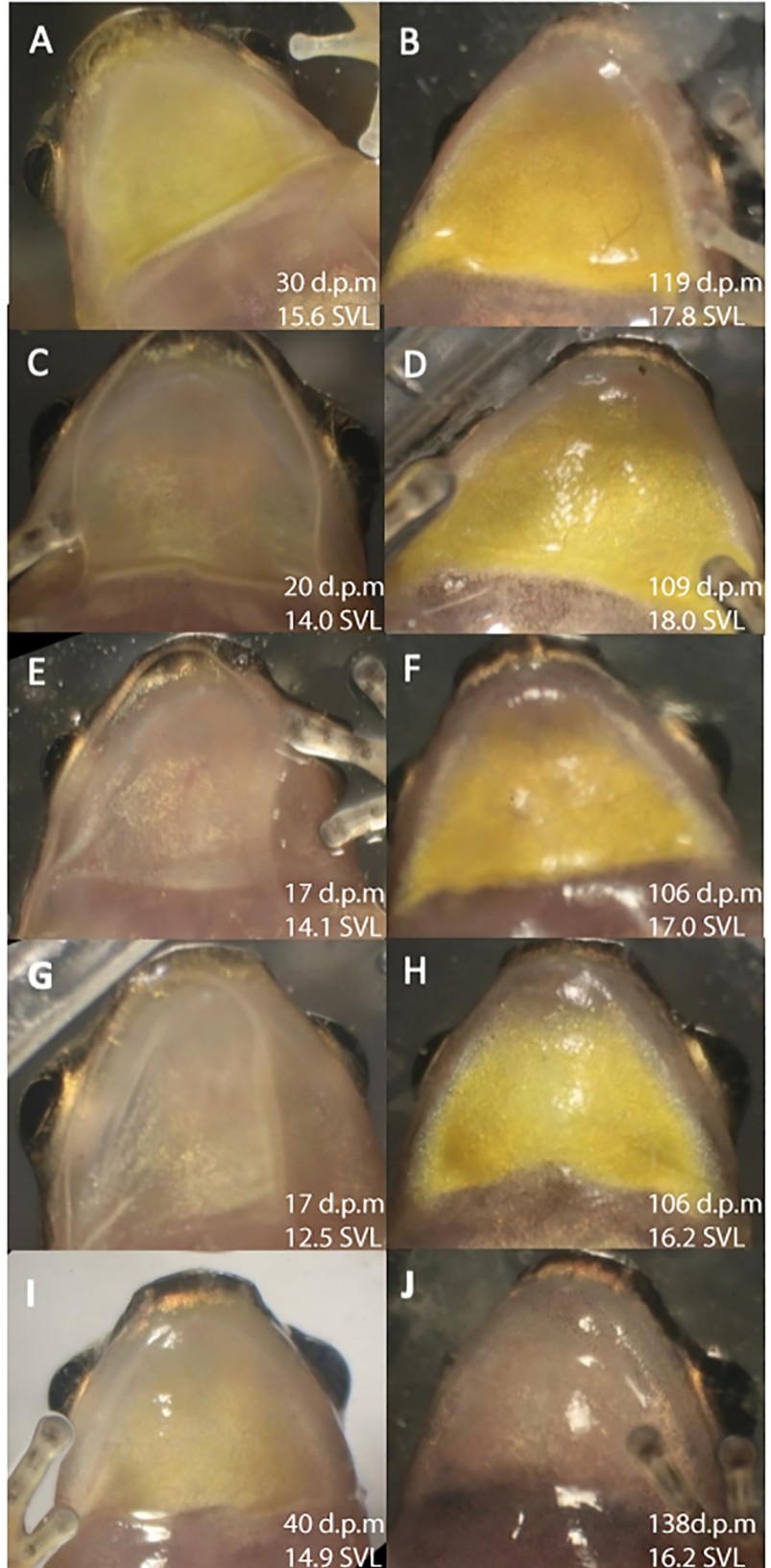

**Fig 4. Throat colour patch development in a selection of post-metamorphic *M. trinitatis* reared in captivity.** d.p. m = days post metamorphosis. Early development left hand column; later stages of the same froglets to the right. Early stages show slight or no yellow pigmentation. Later stages all with yellow throats, female, except frog 5 J, male.

## New field observations on *Mannophryne trinitatis*

We observed *M. trinitatis* sites at dusk and during the hour after sunset. Some male *Mannophryne* remained active, calling, feeding and transporting tadpoles. Although numbers were fewer and calling less frequent than during the day, frogs were out in the open and not hard to find. During a day-time survey, we noticed a male in normal non-calling colouration calling occasionally i.e. not at the usual high frequency. As we watched, this frog began to change to black and over the same time, its calling frequency increased, the full change taking about 20 minutes.

## Sexual dichromatism in *Mannophryne olmonae*

Our field observations confirm that *M. olmonae* males are black when calling. On a late afternoon in June 2014 we first observed a group of three males on a rock by a stream near the Charlotteville- Bloody Bay road (Tobago). All three were calling and all were black. One soon hopped away. The other two called facing one another until one other hopped away, leaving the 'victor' of the encounter (Fig 6A and 6B). We made many similar observations over the next four weeks, but did not actually observe the colour transformation from brownish to black, although we did see a calling male that was mainly brown dorsally and possibly at the start or end of the transition (Fig 6C and 6D). Of 47 calling males observed, 68% were on rocks, 17% on leaves and only 15% in crevices. In a further visit in 2015, we captured 12 adult females and photographed their throats for colour analysis. All had yellow throat patches with a narrow brownish collar and an R:G ratio ranging from 1.03 to 1.104.

## Comparison of sexual dichromatism across the genus *Mannophryne*

Table 3 shows the results of our literature search for evidence of both dynamic sexual dichromatism in males and sex differences in throat colour, assumed to be ontogenetic. We found that in 8 (maybe 9) out of 20 species, dynamic colour change has been documented. In a few cases, colour change in males has not been observed, but this is generally in species where calling has not been seen. We also found that in all cases, females displayed at least some yellow colouration in the throat region. The table includes any information found on the speed of colour change in males, but this has only been reported in a few species so far.

All data used within paper can be found in S3–S5 Tables.

## Discussion

In this paper, focused on the Trinidad stream frog *Mannophryne trinitatis*, with additional observations on the Tobago stream frog *M. olmonae*, and a review of the colour descriptions of the other *Mannophryne* species so far identified, we show that a) dynamic sexual dichromatism

**Table 2. Mean(+/-SD) values for frog size, R:G ratio and measures of escape response.**

| n | SVL (mm) | Weight (g) | R:G | Total escape distance | Mean distance per jump | Min jump | Max jump | Initial jump |
|---|---|---|---|---|---|---|---|---|
| 81 | 21.2 +/- 2.6 | 1.2 +/- 0.4 | 1.2 +/- 0.1 | 85.6 +/- 40.3 | 18.4 +/- 6.2 | 5.1 +/- 5.6 | 35.8 +/- 10.0 | 24.2 +/- 8.0 |

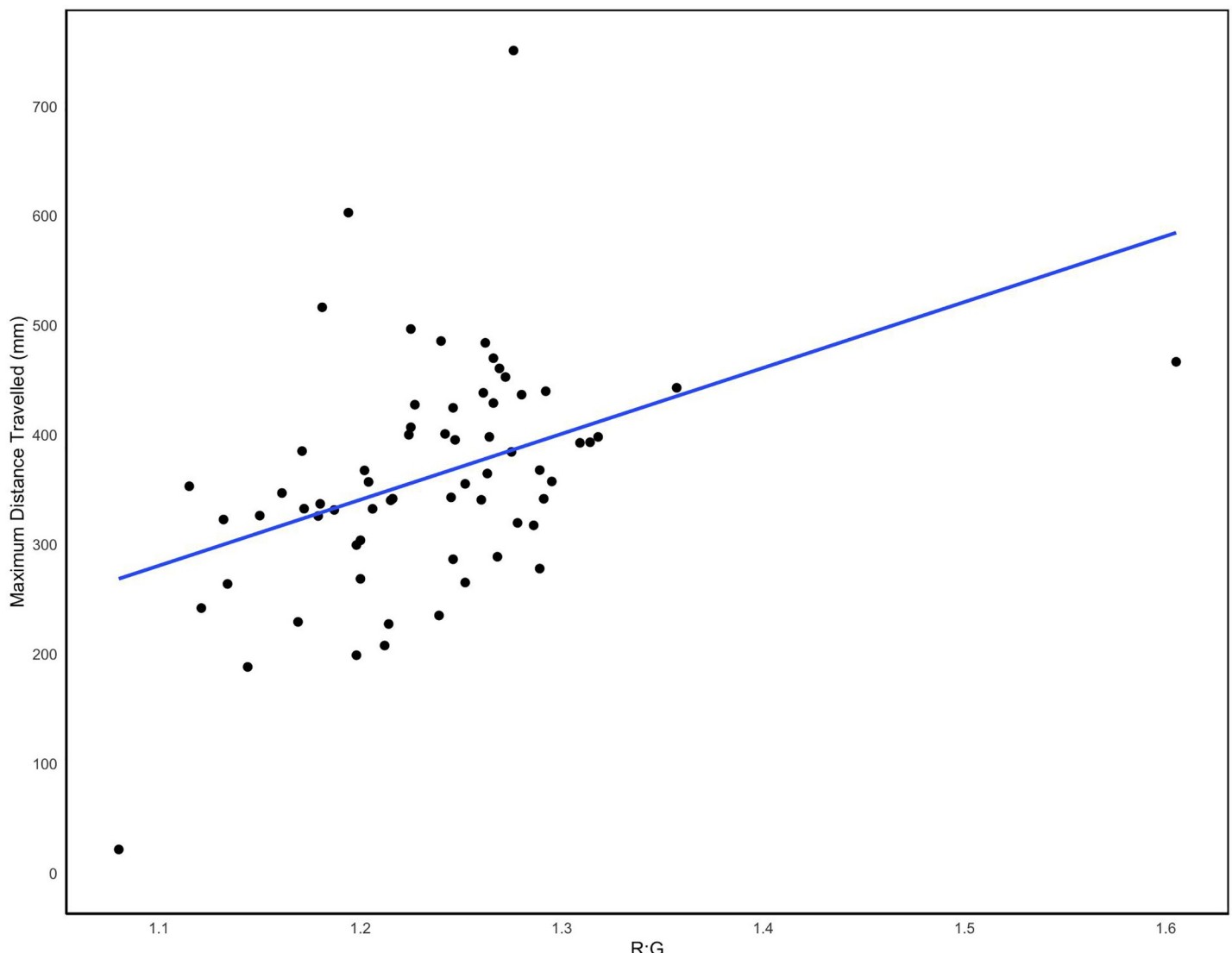

**Fig 5. The relationship between the maximum distance travelled in a single jump (mm) and throat patch colouration (R:G ratio) for *M. trinitatis* females.** The line shows linear regression.

is widespread in the genus, and b) that ontogenetic sexual dichromatism, principally involving the development of a bright yellow throat patch occurs throughout the genus in females. This highlights another example of frogs exhibiting both dichromatisms, this time across a whole genus. We show that the yellow throat patch is highly variable in *M. trinitatis* and *M. olmonae*, and that colour development begins soon after metamorphosis in *M. trinitatis*. We also provide a test that supports our hypothesis that throat colour provides a signal of female quality.

Reviews of the occurrence of conspicuous colouration in frogs [2,3] emphasise two general cases. First, aposematic (warning) signals to other species indicate that these frogs are well protected by toxins. A complication may arise where harmless species evolve to mimic the toxic species, gaining protection without incurring the costs of producing toxins. Second, to protect the frog from drawing the unwelcome attention of predators, the conspicuously coloured element is either temporary or concealed, except from the intended receiver.

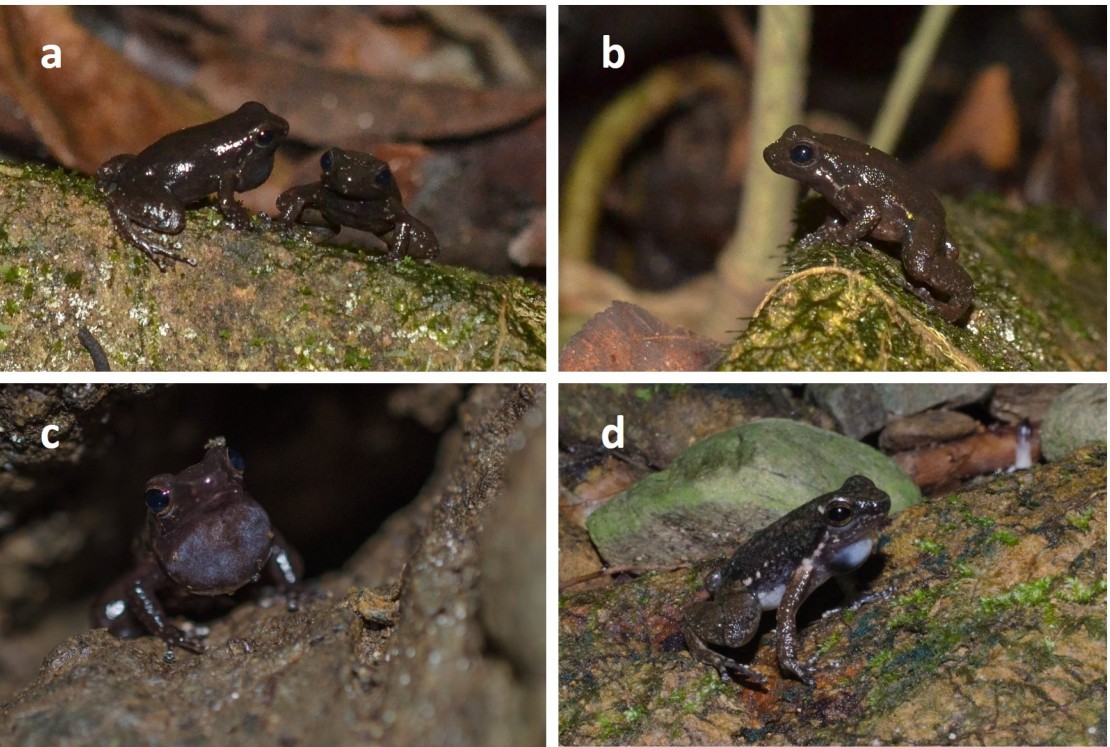

**Fig 6. Males of *M. olmonae*.** (A) two males, jet black all over, soon after a third male had hopped away. (B) the remaining male after the second one in (A) had hopped away; black colour already diminished. (C,D) males with inflated throats, but not yet black all over.

## Dynamic sexual dichromatism in male *Mannophryne*

Bell *et al.* [6] found a relationship between dynamic sexual dichromatism and explosive breeding aggregations in hylids, bufonids and some other groups, and suggested that colour change may assist mate recognition in such situations. However, they also noted that dynamic dichromatism occurs in the absence of breeding aggregations in some species, and related it to intraspecific competition, such as territory defence, in these cases. Colour change in frogs is generally found to be slow (hours to days) and mediated by hormones. However, Kindermann *et al.* [39] found that a dorsal change from brown to yellow in amplexing male *Litoria wilcoxii* took around 5 minutes, and that it could be induced in non-amplexing males by epinephrine injection, implying a neuroendocrine mechanism. Wells [17] reported that male *M. trinitatis* begin calling and then change to black within 1–10 minutes; we also noticed that males start to call before changing to black and that the change occurred over minutes. In other *Mannophryne* species, La Marca [28] reported *M. cordilleriana* changing to black a few seconds after starting to call, and Rojas-Runjaic *et al.* [15] made a similar observation on *M. molinai*. In other species where a change has been seen (8 out of 20 species, plus probably *M. riveroi*: Table 3), the rate of change has not been noted. In a few accounts, the return from black to brown has been reported to be similarly fast. Although no research has been reported on the mechanism of colour change in male *Mannophryne*, the speed implies a neuroendocrine process, and a study to test this is needed.

The function of the male change to black is unclear. *Mannophryne trinitatis* is not an explosive breeder: frogs are distributed along stream sides, with females holding long-term territories. Breeding can occur throughout the long wet season, presumably dependent on females

**Table 3. Reported occurrences of ontogenetic and dynamic dichromatism in living specimens of *Mannophryne* species.**

| Species (sources in brackets) | Males | Females |
|---|---|---|
| *M. caquetio* (1,8) | Dark grey abdomen and throat; no report of change to black when calling | Black, well defined collar; no mention of yellow throat in (1), but (8) shows a female referred to as *M. caquetio* with posterior half of yellow throat |
| *M. collaris* (2) | Throat grey; no report of change to black | Throat yellow; collar black or brown |
| *M. cordilleriana* (2) | Collar dark but inconspicuous; throat dark, chin yellowish tones; they turn black all over a few seconds after calling starts | Anterior part of throat and margins grey; posterior part yellow; black collar with small white flecks |
| *M. herminae* (2,3) | Grey chin and throat; collar solid grey. Skin blackens when calling | Yellow throat and chin; grey collar with silver flecks |
| *M. lamarcai* (4) | Venter, including throat, various shades of grey. Collar distinct in about one third of specimens. Calls produced by black males, but colour change not observed | Throat bright yellow, with solid dark brown collar showing white flecks. Chest yellow |
| *M. larandina* (5) | Males change from brown to black when calling; throat dark grey | Wide brown collar; throat and abdomen yellow |
| *M. leonardoi* (6) | Throat grey; collar grey. Not observed to change collar when calling | Throat and chest yellow; collar thin, well defined and grey |
| *M. molinai* (3) | throat grey; collar diffuse to solid. Skin blackens when calling, reverting quickly to normal when calling ceases | Throat yellow; dark collar, narrow, reticulated to complete |
| *M. neblina* (2) | darker overall than females, with less ventral yellow. No report of change to black | Orange yellow throat; narrow collar, sometimes poorly defined |
| *M. oblitterata* (2) | Completely black ventrally, except for greyish throat. No collar evident. Not observed turning black | Bright yellow throat, tending to orange; chin and lip boarder white. Dark collar with pale flecks; bright yellow throat; black collar |
| *M. olmonae* (2, 7) | Grey throat, black collar. Males turn black all over when calling | Bright yellow throat; black collar |
| *M. orellana* (8) | Throat pale grey anteriorly, dark posteriorly; wide dark collar. Calling observed in field, but change to black not reported | Posterior half of throat yellow, stippled anteriorly with brown; collar dark, narrower than males |
| *M. riveroi* (2,9) | Grey to nearly black throat; wide collar, chocolate to dark grey. Calling males chocolate to nearly black | Bright yellow throat; also yellow on other ventral surfaces. Collar chocolate brown to grey |
| *M. speeri* (10) | Grey stippled ventral surface; no collar. Calls not heard, nor colour change observed | Posterior half of throat yellow; collar wide, dark grey |
| *M. trinitatis* (11) | Grey throat with dark variable collar. Jet black soon after calling starts, the change taking a few minutes | Bright yellow throat, variable in extent; dark variable collar |
| *M. trujillensis* (12) | Throat grey to dark grey, with some posterior yellow spots. Collar wide and dark. Neither calling nor colour change observed | Yellow throat with anterior part grey. Black collar. Yellow more intense in larger females, extending to venter |
| *M. urticans* (3, 8) | Throat grey; collar dark grey, diffuse. Few males heard, and no reports of colour change | Posterior two thirds of throat yellow; narrow dark collar, almost broken medially |
| *M. venezuelensis* (13) | Throat grey; collar poorly defined, relatively unpigmented. Turn black when calling | Throat and sometimes chest yellow; collar poorly defined and relatively unpigmented |
| *M. vulcano* (8) | Throat grey' indistinct black collar. Calling has been observed, but no report of colour change | Yellow throat; pale collar |
| *M. yustizi* (14) | Throat and venter dark grey; no collar. Not seen calling | Throat bright yellow; collar dark brown with white spots |

Code for sources: 1, Mijares–Urrutia and Arends (1999a) [27]; 2, La Marca (1994) [28]; 3, Rojas-Runjaic *et al.* (2018) [15]; 4, Mijares-Urrutia and Arends (1999b) [29]; 5, Yustiz (1991) [30]; 6, Manzanilla *et al.* (2005) [31]; 7, Hardy (1983) [32], Alemu *et al.* (2007) [33] and this paper; 8, Barrio-Amoros *et al.*(2010a) [14]; 9, Barrio-Amoros *et al.* (2010b) [34]; 10, La Marca (2009) [35]; 11, Wells (1980) [17]; 12, Vargas Galarce and La Marca (2006) [36]; 13, Manzanilla *et al.* (2007) [37]; 14, La Marca (1989) [38].

having ripe egg clutches [16,17]. *Mannophryne*, although closely related to dendrobatids, lack toxic protection and are generally cryptic in colouration and behaviour. Their usual habitat is the margins of rocky streams in tropical forest, and they are mainly active during the day. Light levels are low, and the habitat provides abundant shaded crevices where frogs with mottled dorsal colouration of browns, greys and blacks are well concealed. Wells [17] found that males with normal brownish colouration never attacked other males, nor were they attacked by calling males. However, aggressive encounters between black calling males were common. Calling males did not appear to be particularly territorial in their behaviour, often changing

calling site, usually a conspicuous position such as a rock or log (however, we have often seen males calling from shaded crevices). Since both calling and black colouration make the males conspicuous, it is unclear why both signals are needed, especially when they likely increase the risk of predation. Mimicry is unlikely to be at work here. Although some toxic frogs are conspicuously black (for example, dendrobatids of the genus *Ameerega*), their ranges do not appear to overlap with those of *Mannophryne* [40]. The key to the dynamic black signal in males may lie in the unusual territorial behaviour of female *Mannophryne* (see later). In order to attract a female, the males may need to demonstrate their own quality by having visibly successful encounters with other males, or by being conspicuous (colour and sound) for an extended period.

Bell *et al.* [6] checked 13 species of *Mannophryne* for the occurrence of colour change in the males and found that only *M.trinitatis* fitted their criteria, which required photographic evidence. However, we read original species descriptions and some later reports. Since the occurrence of the black colour is transient and not found in preserved specimens, definitive sightings require field observations of calling males. These are often lacking in reports and colour change is often only briefly referred to, since it cannot be used as a species identification criterion in preserved specimens. Of the 12 species reviewed by Bell *et al.* [6] as lacking colour change, we found colour change descriptions in three cases: *Mannophryne herminae*, *M. olmonae* and *M. venezuelensis*. In addition, the seven species not covered by Bell *et al.* yielded three cases of colour change: *M. cordilleriana*, *M. larandina* and *M. molinai*.

If male colour change occurs in some *Mannophryne* but not others, a possible explanation is phylogeny. Manzanilla *et al.* [41] analysed mitochondrial DNA sequences from 13 of the 15 *Mannophryne* then known. They identified three clades of five, one and three species respectively. More recently, Grant *et al.* [13] analysed 14 species and essentially confirmed Manzanilla *et al.*'s clades, though they noted some anomalies in the species so far identified. Male colour change while calling has been reported in species belonging to both of the larger clades. Given that all *Mannophryne* species appear to live in similar habitats with similar behavioural ecology, our hypothesis is that colour change in calling males is likely to occur throughout the genus, and that those species where it has not been reported have not yet been adequately observed in the field.

### Ontogenetic sexual dichromatism in *Mannophryne*

Amongst the anurans, hyperoliids (reed frogs) are the group previously shown to commonly undergo ontogenetic sexual dichromatism, with females developing a bright colour while the males remain dull (found in 35 of 215 species [5,42]). As we have shown, ontogenetic dichromatism in female throat colour appears to occur throughout *Mannophryne*. Females have yellow throat patches of varying size and shade, backed by a dark collar of variable width and tone; the yellow patches sometimes extend beyond the collar on to the abdomen. Males may have a collar, but their throats are invariably grey to black, never yellow. Some species accounts refer to the colour patterns of juveniles, but our study presents the first account of ontogenetic changes in throat colour from the end of metamorphosis to near maturity, and it is clear that the adult sexual dichromatism is ontogenetic. Wells [17] found that female *M. trinitatis* aggressively defend their territories and that the signal used to denote a territory holder is pulsation of the throat with the head held high so that the yellow throat patch, pulsation and dark collar are clearly visible to any approaching conspecific. Durant and Dole [43] described similar behaviour in female *Colostethus* (now *Mannophryne*) *collaris*. Exposure of a coloured signal that is normally concealed has been infrequently reported in frogs. One other example is the

brightly coloured foot webbing in the foot-flagging frog *Staurois parvus*, where males extend and rotate their legs, displaying the colours, during social interactions [44].

To our knowledge, no one has previously suggested that variation in the yellow colour acts as a signal of female quality. However, across the animal kingdom, bright yellow patches are often used in this way, related to the cost of synthesising the carotenoids on which yellow colours are often based, and to the role of carotenoids in immune system function [18]. In anurans, a relevant example is the orange-coloured (carotenoid-based) vocal sac of chorusing tree frogs, where females prefer males with colourful compared to pale vocal sacs [19,45].

Our data show considerable individual differences in female throat colour in both *M. trinitatis* and *M. olmonae*. To test whether throat colour differences act as signals of female quality, it would be best to recover the winners and losers after territorial encounters and measure their throat colours. However, in practice, the combination of the need to observe from a distance (to prevent disturbance) and the nature of the terrain (rocks with abundant deep crevices used by frogs for concealment) made this unfeasible. We used a proxy for quality instead. Royan *et al.* [46] reported experiments where the escape response trajectories of captured *M. trinitatis* were measured using an outdoor arena. They found that angle of escape was variable, indicating a degree of unpredictability, which could help individuals to escape potential predators. We reasoned that differences in escape response might provide as good a measure of quality as the results of territorial encounters. Our results showed a positive relationship between R:G ratio and two measures of escape performance, providing evidence that throat patch colour is indeed a signal of quality. However, the signal emitted by females is more than simply the colour of the throat. Our measurements over a large population of female *M. trinitatis* and a small sample of *M. olmonae* show variation in throat colour, patch size, the width and colour of the collar, and the extent of the yellow patch posterior to the collar so the quality of the signal may include all these components.

## Conclusion and a hypothesis

The adaptive significance of female sexual dichromatism in *Mannophryne* seems clear: it is associated with territorial defence. Territorial behaviour in dendrobatids is common, but mainly involves males [2]. Long-term defence of a territory by female frogs, as occurs in *Mannophryne* (extrapolating from the species where it has been demonstrated, such as *M. trinitatis* and *M. collaris*), is very rare. The obvious suggestion is that females are defending resources, most likely food. Frogs captured in late afternoon showed that females had significantly fuller stomachs (small insects, arachnids and occasional snails) than males [47]. It is not known whether food resources are patchily distributed or simply related to area: a study on territory size in relation to throat colour would be helpful. If we are correct in concluding that throat colour differences in females provide a measure of quality, then it is likely that the throat pulsing observed by Wells [17] during female to female intraspecific confrontation provides a signal, potentially avoiding a physical confrontation. Additionally, it would be beneficial to the males to choose the best possible quality of mate, demonstrated by their visually striking territorial defence. Females also need to select the best possible mate: males guard the eggs throughout incubation and then transport the hatchlings to a suitable body of water. Downie *et al.* [48] found that transporting males may take several days to locate a suitable pool, ideally one lacking predators. This post-hatching transportation phase probably makes it unfeasible for male *Mannophryne* to guard multiple egg clutches, as occurs in some other clutch-guarders, such as glass frogs [49]. Males therefore have to be sure that the quality of the eggs is high enough to justify the considerable investment in time involved in incubation and transportation. Males can demonstrate their quality by calling for long periods, and by turning

conspicuously black, both hazardous and possibly energetically expensive activities. The mating dances the males perform also contribute [17]. Our hypothesis therefore is that the occurrence of sexual dichromatism in both sexes derives from the resource-based territoriality of the females, using throat colouration as a signal in female to female conspecific confrontations, and strong selection for quality in both sexes. Males avoid the predation costs of conspicuousness by their colour signal being temporary and quickly turned off and on; for females, the colour signal is concealed, except when used against conspecific receivers. Follow up studies on the reproductive success, or even just territory defensive success, of females with different throat colouration would be very interesting. However, we are aware of how difficult this would be. Further studies on the development of the female colouration across multiple species and over a longer time period may lead to some interesting results. Clarification of the presence or absence of the dynamic colour change for males of species from the rest of the genus would also be useful.

## Supporting information

**S1 Table. Frog collection data.** Data for the number of individuals sampled from each site per year.
(DOCX)

**S2 Table. Significance values from analysis of escape response trials.** Breakdown of the significance values from each statistical test conducted.
(DOCX)

**S3 Table. Data from colour variability in female *Mannophryne trinitatis* section.** All data collected and used for the colour variability in female *Mannophyne trinitatis* study.
(XLS)

**S4 Table. Data from colour development in juvenile *Mannophryne trinitatis*.** All data collected and used for analysis in colour development in juvenile *Mannophryne trinitatis* study.
(DOCX)

**S5 Table. Data from escape responses in *Mannophryne trinitatis* in relation to female throat colour.** All data collected and used for analysis in escape responses in *Mannophryne trinitatis* in relation to female throat colour study.
(XLSX)

## Acknowledgments

The fieldwork upon which the results reported here are based was carried out on University of Glasgow expeditions to Trinidad, and to Tobago over the years 2015–18, with many students helping make observations and catch frogs. Fieldwork permits were provided in Trinidad by the Government's Wildlife Section, Forestry Division, in St. Joseph. In addition, Special Export License 001192 (29/6/16) gave permission for export of the tadpoles used in the rearing experiment. Thanks to Anna Bandoo and Romano McFarlane for facilitating these permits. In Tobago, Angela Ramsay of the Tobago House of Assembly's Department of Natural Resources and the Environment provided fieldwork permits. Multiple agencies supported the expeditions financially, notably the University of Glasgow's Chancellor's Fund, Glasgow Natural History Society, the Gilchrist Educational Trust, the Thriplow Trust, and Dennis Curry's Charitable Trust. A Percy Sladen Memorial Fund grant funded the camera used in the escape response experiments. The University of the West Indies and the Asa Wright Nature Centre provided laboratory accommodation in Trinidad, and Pat Turpin in Tobago. We thank Dr James D.

Burgon for assistance on the technical and conceptual aspects. We thank Rayna C. Bell for her comments on an earlier version of this paper.

## Author Contributions

**Conceptualization:** Mark S. Greener, Christopher J. Pollock, Annabeth Wilson, Chun Yin Lam, J. Roger Downie.

**Data curation:** Mark S. Greener, Christopher J. Pollock, J. Roger Downie.

**Formal analysis:** Mark S. Greener, Emily Hutton, Christopher J. Pollock, Annabeth Wilson, Chun Yin Lam, Mohsen Nokhbatolfoghahai, Michael J. Jowers, J. Roger Downie.

**Investigation:** Mark S. Greener, Emily Hutton, Mohsen Nokhbatolfoghahai.

**Methodology:** Mark S. Greener, Emily Hutton, Christopher J. Pollock, Annabeth Wilson, Chun Yin Lam, Mohsen Nokhbatolfoghahai, J. Roger Downie.

**Project administration:** Mark S. Greener, J. Roger Downie.

**Supervision:** J. Roger Downie.

**Visualization:** Mark S. Greener, Christopher J. Pollock, Mohsen Nokhbatolfoghahai.

**Writing – original draft:** Mark S. Greener, Emily Hutton, Christopher J. Pollock, Michael J. Jowers, J. Roger Downie.

**Writing – review & editing:** Mark S. Greener, Emily Hutton, J. Roger Downie.

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
