## [Decision Letter · Decision Letter 0]

2 Jan 2020

PONE-D-19-21985

Sexual dichromatism in the neotropical genus Mannophryne (Anura: Aromobatidae)

PLOS ONE

Dear Mr Greener,

Thank you for submitting your manuscript to PLOS ONE. After careful consideration, we feel that it has merit but does not fully meet PLOS ONE’s publication criteria as it currently stands. Therefore, we invite you to submit a revised version of the manuscript that addresses the points raised during the review process.

Two expert individuals in the field have reviewed the manuscript. Both expressed enthusiasm for the studies and the general findings. However, both expressed some concerns regarding the lack of details in the methods, need to reframe the Intro and Discussion, including citing relevant studies, and the need for better statistical analyses.

We would appreciate receiving your revised manuscript by Feb 16 2020 11:59PM. To enhance the reproducibility of your results, we recommend that if applicable you deposit your laboratory protocols in protocols.io, where a protocol can be assigned its own identifier (DOI) such that it can be cited independently in the future. For instructions see: http://journals.plos.org/plosone/s/submission-guidelines#loc-laboratory-protocols

We look forward to receiving your revised manuscript.

Kind regards,

Cheryl S. Rosenfeld, DVM, PhD

Academic Editor

PLOS ONE

Journal Requirements:

1. 

2.  Thank you for providing additional information regarding the field permit and export licence obtained for your study. However, we feel that the text in your manuscript is still not fully clear as to which aspects of your study were covered by which permits. At this time, please could you amend the text of your Methods/Ethics Statement to clarify whether the studies performed at UWI and Simla were covered by the field research permit, and provide any relevant permit reference numbers? Thank you for your attention to this further query.

3.  Thank you for including your ethics statement:  "Ethics Statement - N/A

Field Research -Government’s Wildlife Section, Special Export License 001192 (29/6/16)".   

To comply with PLOS ONE submissions requirements, please provide the following information in the Methods section of the manuscript and in the “Ethics Statement” field of the submission form (via “Edit Submission”):  

*  Please indicate whether an animal research ethics committee prospectively approved this research or granted a formal waiver of ethics approval.*  Please enter the name of your Institutional Animal Care and Use Committee (IACUC) or other relevant ethics board. Also include an approval number if one was obtained.

*   If anesthesia, euthanasia, or any kind of animal sacrifice is part of the study, please include briefly in your statement which substances and/or methods were applied.

For additional information about PLOS ONE submissions requirements for ethics oversight of animal work, please refer to http://journals.plos.org/plosone/s/submission-guidelines#loc-animal-research  

For additional information about PLOS ONE submissions requirements for animal ethics, please refer to http://journals.plos.org/plosone/s/submission-guidelines#loc-animal-research

Please provide an amended Funding Statement that declares *all* the funding or sources of support received during this specific study (whether external or internal to your organization) as detailed online in our guide for authors at http://journals.plos.org/plosone/s/submit-now.  

Please state what role the funders took in the study.  If any authors received a salary from any of your funders, please state which authors and which funder. If the funders had no role, please state: "The funders had no role in study design, data collection and analysis, decision to publish, or preparation of the manuscript."

Reviewers' comments:

Reviewer's Responses to Questions

**Comments to the Author**

1. Is the manuscript technically sound, and do the data support the conclusions?

Reviewer #1: Partly

Reviewer #2: Yes

2. Has the statistical analysis been performed appropriately and rigorously? 

Reviewer #1: No

Reviewer #2: Yes

3. Have the authors made all data underlying the findings in their manuscript fully available?

Reviewer #1: Yes

Reviewer #2: Yes

4. Is the manuscript presented in an intelligible fashion and written in standard English?

Reviewer #1: Yes

Reviewer #2: Yes

5. Review Comments to the Author

Reviewer #1: I found there a lot to be interested in here both empirically (the new data presented) as well as the information reviewed and summarized from the literature. However, the “package” needs better selling. The authors give good reasons for the work they did to demonstrate dynamic and ontogenetic dichromatism but less of a good job justifying the data they collected on variation in throat color, escape performance, etc. I think there is a very interesting story in here and a lot of work has been done to uncover that story; I think there just needs to be better packaging in the Intro and Discussion to really make it clear to the reader why these questions need answering and how they are all related to one another. Below are detailed comments.

Detailed Comments

Line 84 “tree frog example”, does this need to be here?

Line 98-99 For the uninitiated, it may be beneficial to define Bd as a fungal pathogen.

Lines 119-120 I wonder if using Generalized Linear Mixed Models (GLMMs) might be a more powerful and flexible approach to take here. The main advantage of GLMMs is that they can be used with data that are not continuous and/or do not conform to the standard parametric assumptions.

Line 145 Just one ‘l’ in caliper or is this an American/British English thing? (also 156)

Lines 177 There might be some multiple comparisons issues here with the various measures of escape performance. It seems likely that these are not independent of one another but if one uses a alpha of 0.05 for every analysis performed then they are be treated that way. Perhaps a correction for multiple comparisons is necessary. See Table S2.

Line 181-183 Is this section needed?

Line 193 Checked what?

Lines 269-287 Why no graphs illustrating the escape performance test results?

Lines 299-309 this seems more like background information but it is placed in the Results section.

Line 340 focussed

Lines 332-339 This seems repetitive from the Intro; maybe just start at line 340.

Line 390 Perhaps briefly describe how you envision Zahavi’s principle being at work here.

Lines 445-47 Is escape performance necessarily related to mate attraction and reproductive success? It is a proxy, but is it a good proxy for who wins and who loses in territorial encounters? Maybe larger females have larger R:G ratios (on average, demonstrated in Fig. 2) and larger females also can jump farther (stated in line 287-289). Maybe jump performance only is driven by size (but since size also influences color, it makes it appear as though this is also important. A statistical analysis that clearly separates out the influence of SVL versus color on jump performance is needed. And I think they authors already attempted it, with the two-way ANOVA referred to in line 289 but the full results of this or another test (the influence of color on jump performance after the effect of SVL has been removed) is needed. Maybe an ANCOVA with SVL as the covariate?

Line 475 Perhaps suggesting some effective tests of the proposed hypothesis here. What would be really nice is to have data on actual reproductive success of females with different throat colors, though I realize how difficult this would be to get in the real world.

Table 2 Use n instead of samples and other abbreviations to tighten up this table a bit.

Reviewer #2: This is a study of coloration in Mannyphryne frogs from Trinidad and Tobago. The study 1) analyzes a large data set of photographs of female M. trinitatis to document variation in yellow throat coloration and relate this to body size, 2) documents sex-specific throat color development, 3) supports the hypothesis that throat coloration is related to quality by showing females with yellower throats have farther escape jumps, 4) documents sexual dichromatism in a close relative, M. olmonae, and 5) reviews literature which suggest that ontogenetic and dynamic dichromatism are widespread in this genus.

In general, I think this is a really valuable study, with nice, big sample sizes and sampling across many populations. I also really like the escape response experiments to link this variation in coloration with some measure of quality or condition. The behavioral experiments and the across-site sampling are strengths to the study.

I have no major concerns other than wanting more details about the photography protocol and how the authors ensured even lighting across photographs. I think this is simply an issue of more detail being needed in the methods. Otherwise, I have a bunch of minor comments where I think more details are needed or things could be clarified. I’m confident the authors can address these concerns.

Line 34: Suggestion: Delete “It is well established that” from the beginning of this sentence. And include a general reference for this first sentence. Wells (2007) or Gerhardt and Huber (2002) are good ones.

Gerhardt, H. C., and Huber, F. (2002). Acoustic Commmunication in Insects and Anurans. Chicago: University of Chicago Press.

Wells, K. D. (2007). The Ecology and Behavior of Amphibians. Chicago: University of Chicago Press.

Line 35: should be “acoustic signals”

Lines 47-48: A recent paper has documented both in an Aromobatid frog, Anomaloglossus beebei (Engelbrecht‐Wiggans and Tumulty, 2019).

Engelbrecht‐Wiggans, E., and Tumulty, J. P. (2019). “Reverse” sexual dichromatism in a Neotropical frog. Ethology. doi:10.1111/eth.12942.

Lines 50-50: I recommend breaking up this sentence, it is kind of hard to follow the logic here. If a color signal is not useful for nocturnal breeders, why later in the sentence do you say that it is?

Lines 49-53: Probably worth also discussing the potential for temporary color change to function as a sex recognition signal for explosive breeders. This seems to be the case in some species (e.g., Sztatecsny et al., 2012) and this hypothesis is support by the comparative analysis of (Bell et al., 2017).

Sztatecsny, M., Preininger, D., Freudmann, A., Loretto, M. C., Maier, F., and Hödl, W. (2012). Don’t get the blues: Conspicuous nuptial colouration of male moor frogs (Rana arvalis) supports visual mate recognition during scramble competition in large breeding aggregations. Behav. Ecol. Sociobiol. 66, 1587–1593. doi:10.1007/s00265-012-1412-6.

Bell, R. C., Webster, G. N., and Whiting, M. J. (2017). Breeding biology and the evolution of dynamic sexual dichromatism in frogs. J. Evol. Biol. 30, 2104–2115. doi:10.1111/jeb.13170.

Line 86: Please articulate the hypothesis of this experiment. Also, I don’t think “fitness” is the right word here. I think this is more a question of whether the signals indicate something about “condition” or “quality”. If acquiring and putting carotenoids into skin is somehow costly, then individuals in better condition may be able to invest more of the limited carotenoids to coloration. Escape response could be a metric of condition or quality, but I think it’s a step too far to say it is linked with fitness (e.g., maybe bolder individuals have higher fitness and are less likely to escape?). Whether having increased yellow coloration is linked with fitness is another question that will take a few more pieces of evidence to link to. So, I recommend being more cautious with word choice here.

Line 98: Please provide slightly more detail on “returned to their collection sites”. Approximately how large is a “site”? Poison frogs can sometimes have very small home ranges, so, ensuring that the frogs were within ~10-20m of their capture points or putting each frog back at its exact capture location is very different than dumping all the frogs back in the same general area where they were caught.

Lines 105-114: I’m a bit unclear on the photography protocol. Were Colorchecker color standards included in each photographs or photographed separately? If separately, the lighting conditions need to be identical across photographs. Please provide more details on what a color profile is and how it ensures standard lighting conditions across photographs.

Line 114: What was the areas of interest? Please be precise.

Lines 114-115: Why only R and G, and not B? Also please provide some justification for using the R:B ratio as your measurement.

Lines 115-117: I’m confused by this sentence, why does combining R:G and SVL allow for comparisons between sites?

Lines 121-122: How was presence of colour posterior to the collar measured? This is the first we are hearing about this in the methods.

Lines 126-127: The fact that 60 tadpoles were collected suggests that these tadpoles were from, at least, five or more different families (assuming a clutch size of around 10-12 eggs?), but please provide some justification for this. If they were all collected from one small pool, how representative is this sample of the population as a whole? Do many different parents deposit tadpoles in the same pool?

Lines 146-147: Please try to be more precise here. The four-week intervals part is confusing me. Based on this sentence it sounds like the froglets were photographed approximately once every 4 weeks for approximately 16 weeks, is this right?

Line 157: Check grammar here, missing subject?

Line 158: If you give a time for night photography, you should also give one for morning photography.

Line 159: Please describe lighting conditions. This is especially important to be able to compare the results of this photography to the descriptive statistics of the throat color variation earlier.

Line 162: Need comma after “ratio”?

Lines 181-183: Please provide a bit more detail on observations. For example, did you conduct focal observations of individuals in a systematic way or only if you observed something interesting or unusual?

Line 192: Perhaps change this to “We performed a literature review on…”

Figure 3. Need clarification on what, exactly, the y-axis is showing. What is “SVLxR:G” and how was it calculated?

Lines 224-229: Unclear if these are all quantitative results or sometimes qualitative. I think this relates to my earlier question about what the regions of interest were. I was surprised to read hear about quantitative results about presence of yellow posterior to the color because I didn’t see anything about measuring this in the methods section.

Figure 5. Same issue of unclear y-axis as figure 3. Also, is “SVLxR:G” the same thing as “SVL*R:G”? If so, why use different characters for the interaction? If not, need a lot more clarification on what is being measured. This figure legend also needs details on the Collar Yes vs. No key. No explanation for this key is provided.

Line 245: Recommend changing “classed” to “classified”.

Lines 252-254: I think these results are really interesting (i.e., that yellow coloration shows up prior to sexual maturation) and I would appreciate seeing more data to support this conclusion. It would appear that you probably have data on adult size and size of juveniles through development. Perhaps some sort of graph showing SVL on the y-axis over time on the x-axis, and a line or range showing the SVL/time when yellow coloration first shows up?

Figure 6. Could you add the details about age and SVL to the figure itself? Perhaps as small inset details on the bottom of each image? This would make it a lot easier on readers instead of having to look back and forth between the figure and the figure legend for this information.

Table 2. I don’t think this table is all that informative given the goal of the experiment. Are we really that interest in the overall mean? Why not show graphs of the relationships of interest? i.e., the relationships between R:G ratio and measures of jump performance?

Lines 300-309: I’m confused why this background information is presented in the results. Shouldn’t it be in the Introduction?

Lines 317-320: Could you add photos of the female throat of M. olmonae too? Might be a nice addition.

Lines 324-30: I really think this Table S3 should be in the main text and not the supplementary. Otherwise, why even devote a results subsection to it? And I also think more summary details should be included in this results sub-section. For example, for how many species has female yellow throat coloration been observed? Or male color change?

Line 340: “focused”

Lines 340-346: Please also provide a clear statement that your results supported the hypothesis that female throat coloration is related to female quality (i.e., yellower females jumped farther).

Lines 332-346: A suggestion: I think this discussion could be a lot stronger if you provide clearer statements of what your study accomplished and what the key results were right in the beginning, without making the reader search for it later on.

Line 365: La Marca reference?

Line 390. Seems odd to suggest Zahavi as an afterthought with no real explanation. If this is a reasonable hypothesis, please provide a more clear explanation to how turning black when calling could be a “handicap”.

Line 398. Please include the full genus name before the first Mannyphryne species listed, and “M.” before each subsequent species of the same genus.

Lines 401-407: I find this explanation based on phylogeny unclear. Are you saying that based on the presence of this trait among clades that is likely ancestral to the genus?

Lines 412-414: Long sentence. I recommend moving reed frogs to the beginning so it doesn’t take so long to get to the subject.

Line 429: Change to “no one”

Lines 465-467: Is this true? Allobates femoralis have been studied in quite some detail and males typically have several clutches in their territories and may take 1-2 days to transport tadpoles (round trip), if I recall correctly.

Lines 451-475: I was a bit surprised to hear the only concluding hypothesis is one about mutual mate choice (i.e., suggesting males may choose females based on yellow throat coloration). Didn’t Wells observe contests between females that seemed to be resolved with these yellow throat signals? Given these natural history observations, isn’t intrasexual communication of dominance associated with territory defense also a reasonable hypothesis for the function of these signals?

Table S1. Please specify that the data presented are sample sizes (assuming that’s what they are).

6. PLOS authors have the option to publish the peer review history of their article (what does this mean?). If published, this will include your full peer review and any attached files.

Reviewer #1: No

Reviewer #2: Yes: James Tumulty

---

## [Author Response · Author response to Decision Letter 0]

16 Feb 2020

5. Review Comments to the Author

Reviewer #1: I found there a lot to be interested in here both empirically (the new data presented) as well as the information reviewed and summarized from the literature. However, the “package” needs better selling. The authors give good reasons for the work they did to demonstrate dynamic and ontogenetic dichromatism but less of a good job justifying the data they collected on variation in throat color, escape performance, etc. I think there is a very interesting story in here and a lot of work has been done to uncover that story; I think there just needs to be better packaging in the Intro and Discussion to really make it clear to the reader why these questions need answering and how they are all related to one another. Below are detailed comments.

Detailed Comments

Line 84 “tree frog example”, does this need to be here?

We feel this example is justified

Line 98-99 For the uninitiated, it may be beneficial to define Bd as a fungal pathogen.

Clarified

Lines 119-120 I wonder if using Generalized Linear Mixed Models (GLMMs) might be a more powerful and flexible approach to take here. The main advantage of GLMMs is that they can be used with data that are not continuous and/or do not conform to the standard parametric assumptions.

We took this advice and conducted GLMMs for this section of the analysis.

Line 145 Just one ‘l’ in caliper or is this an American/British English thing? (also 156)

British spelling has two ‘l’s e.g. callipers

Lines 177 There might be some multiple comparisons issues here with the various measures of escape performance. It seems likely that these are not independent of one another but if one uses a alpha of 0.05 for every analysis performed then they are be treated that way. Perhaps a correction for multiple comparisons is necessary. 

See Table S2.

Benjamini-Hochberg procedure used and alpha adjusted to 0.01. Statistical findings adjusted accordingly. (Line 207)

Line 181-183 Is this section needed?

We thought it better to have a small section in the methods so the section in the results was not ‘out of the blue’.

Line 193 Checked what?

Clarified

Lines 269-287 Why no graphs illustrating the escape performance test results?

Graph showing relationship between R:G ratio and maximum distance travelled added. (Line 311)

Lines 299-309 this seems more like background information but it is placed in the Results section.

Agreed and removed

Line 340 focussed

Technically UK English has the double ‘s’. However, this was changed to avoid confusion.

Lines 332-339 This seems repetitive from the Intro; maybe just start at line 340.

Agreed and removed

Line 390 Perhaps briefly describe how you envision Zahavi’s principle being at work here. 

Removed as this may have been somewhat of a jump

Lines 445-47 Is escape performance necessarily related to mate attraction and reproductive success? It is a proxy, but is it a good proxy for who wins and who loses in territorial encounters? Maybe larger females have larger R:G ratios (on average, demonstrated in Fig. 2) and larger females also can jump farther (stated in line 287-289). Maybe jump performance only is driven by size (but since size also influences color, it makes it appear as though this is also important. A statistical analysis that clearly separates out the influence of SVL versus color on jump performance is needed. And I think they authors already attempted it, with the two-way ANOVA referred to in line 289 but the full results of this or another test (the influence of color on jump performance after the effect of SVL has been removed) is needed. Maybe an ANCOVA with SVL as the covariate?

More appropriate models were fitted to correct the statistical analysis (as per other comments). Larger females have larger R:G ratios, and females with larger R:G ratios jump further, but SVL does not significantly impact escape performance. An ANCOVA analysis of this relationship is no longer needed.

Line 475 Perhaps suggesting some effective tests of the proposed hypothesis here. What would be really nice is to have data on actual reproductive success of females with different throat colors, though I realize how difficult this would be to get in the real world.

Some follow up studies have been suggested

Table 2 Use n instead of samples and other abbreviations to tighten up this table a bit.

Tidied up a bit

Reviewer #2: This is a study of coloration in Mannyphryne frogs from Trinidad and Tobago. The study 1) analyzes a large data set of photographs of female M. trinitatis to document variation in yellow throat coloration and relate this to body size, 2) documents sex-specific throat color development, 3) supports the hypothesis that throat coloration is related to quality by showing females with yellower throats have farther escape jumps, 4) documents sexual dichromatism in a close relative, M. olmonae, and 5) reviews literature which suggest that ontogenetic and dynamic dichromatism are widespread in this genus.

In general, I think this is a really valuable study, with nice, big sample sizes and sampling across many populations. I also really like the escape response experiments to link this variation in coloration with some measure of quality or condition. The behavioral experiments and the across-site sampling are strengths to the study.

I have no major concerns other than wanting more details about the photography protocol and how the authors ensured even lighting across photographs. I think this is simply an issue of more detail being needed in the methods. Otherwise, I have a bunch of minor comments where I think more details are needed or things could be clarified. I’m confident the authors can address these concerns.

Line 34: Suggestion: Delete “It is well established that” from the beginning of this sentence. And include a general reference for this first sentence. Wells (2007) or Gerhardt and Huber (2002) are good ones.

Gerhardt, H. C., and Huber, F. (2002). Acoustic Commmunication in Insects and Anurans. Chicago: University of Chicago Press.

Wells, K. D. (2007). The Ecology and Behavior of Amphibians. Chicago: University of Chicago Press.

Removed start of sentence and added ‘Wells (2007)’.

Line 35: should be “acoustic signals”

Changed

Lines 47-48: A recent paper has documented both in an Aromobatid frog, Anomaloglossus beebei (Engelbrecht‐Wiggans and Tumulty, 2019).

Engelbrecht‐Wiggans, E., and Tumulty, J. P. (2019). “Reverse” sexual dichromatism in a Neotropical frog. Ethology. doi:10.1111/eth.12942.

Added and adapted intro to reflect this recent publication

Lines 50-50: I recommend breaking up this sentence, it is kind of hard to follow the logic here. If a color signal is not useful for nocturnal breeders, why later in the sentence do you say that it is?

Sentences rewritten to be clearer and separated.

Lines 49-53: Probably worth also discussing the potential for temporary color change to function as a sex recognition signal for explosive breeders. This seems to be the case in some species (e.g., Sztatecsny et al., 2012) and this hypothesis is support by the comparative analysis of (Bell et al., 2017).

Sztatecsny, M., Preininger, D., Freudmann, A., Loretto, M. C., Maier, F., and Hödl, W. (2012). Don’t get the blues: Conspicuous nuptial colouration of male moor frogs (Rana arvalis) supports visual mate recognition during scramble competition in large breeding aggregations. Behav. Ecol. Sociobiol. 66, 1587–1593. doi:10.1007/s00265-012-1412-6.

Bell, R. C., Webster, G. N., and Whiting, M. J. (2017). Breeding biology and the evolution of dynamic sexual dichromatism in frogs. J. Evol. Biol. 30, 2104–2115. doi:10.1111/jeb.13170.

Added to the section

Line 86: Please articulate the hypothesis of this experiment. Also, I don’t think “fitness” is the right word here. I think this is more a question of whether the signals indicate something about “condition” or “quality”. If acquiring and putting carotenoids into skin is somehow costly, then individuals in better condition may be able to invest more of the limited carotenoids to coloration. Escape response could be a metric of condition or quality, but I think it’s a step too far to say it is linked with fitness (e.g., maybe bolder individuals have higher fitness and are less likely to escape?). Whether having increased yellow coloration is linked with fitness is another question that will take a few more pieces of evidence to link to. So, I recommend being more cautious with word choice here.

Agreed and changed to quality 

Line 98: Please provide slightly more detail on “returned to their collection sites”. Approximately how large is a “site”? Poison frogs can sometimes have very small home ranges, so, ensuring that the frogs were within ~10-20m of their capture points or putting each frog back at its exact capture location is very different than dumping all the frogs back in the same general area where they were caught.

Added some clarification

Lines 105-114: I’m a bit unclear on the photography protocol. Were Colorchecker color standards included in each photographs or photographed separately? If separately, the lighting conditions need to be identical across photographs. Please provide more details on what a color profile is and how it ensures standard lighting conditions across photographs.

More detail has been added and hopefully this has addressed anything that was unclear

Line 114: What was the areas of interest? Please be precise.

Clarified

Lines 114-115: Why only R and G, and not B? Also please provide some justification for using the R:B ratio as your measurement.

We did not see the need to include the Blue into the ratios, as we were doing a study on the yellow colouration. Considering that Red and Green have a strong link to yellow colouration creation.

Lines 115-117: I’m confused by this sentence, why does combining R:G and SVL allow for comparisons between sites?

After using GLMMs for this section, this is not longer an issue

Lines 121-122: How was presence of colour posterior to the collar measured? This is the first we are hearing about this in the methods.

Clarrified. This was a qualitative “Yes or No” situation.

Lines 126-127: The fact that 60 tadpoles were collected suggests that these tadpoles were from, at least, five or more different families (assuming a clutch size of around 10-12 eggs?), but please provide some justification for this. If they were all collected from one small pool, how representative is this sample of the population as a whole? Do many different parents deposit tadpoles in the same pool?

This was a sample of 60 from a larger number of potential tadpoles in the pond. Therefore, our sample will have offspring from multiple parents.

Lines 146-147: Please try to be more precise here. The four-week intervals part is confusing me. Based on this sentence it sounds like the froglets were photographed approximately once every 4 weeks for approximately 16 weeks, is this right?

Added clarification

Line 157: Check grammar here, missing subject?

Corrected

Line 158: If you give a time for night photography, you should also give one for morning photography.

Time added

Line 159: Please describe lighting conditions. This is especially important to be able to compare the results of this photography to the descriptive statistics of the throat color variation earlier.

Lighting conditions explained

Line 162: Need comma after “ratio”?

Corrected

Lines 181-183: Please provide a bit more detail on observations. For example, did you conduct focal observations of individuals in a systematic way or only if you observed something interesting or unusual?

Clarified. We made occasional interesting observations whilst conducing population surveys in years previous to this study for another paper.

Line 192: Perhaps change this to “We performed a literature review on…”

Changed

Figure 3. Need clarification on what, exactly, the y-axis is showing. What is “SVLxR:G” and how was it calculated?

Statistical method has been changed after the advice of the 1st reviewer. So, this figure has been removed.

Lines 224-229: Unclear if these are all quantitative results or sometimes qualitative. I think this relates to my earlier question about what the regions of interest were. I was surprised to read hear about quantitative results about presence of yellow posterior to the color because I didn’t see anything about measuring this in the methods section.

I have tried to make this clearer. The observations were qualitative in nature and apart from the presence of colour under the collar (mentioned previously) we are only mentioning them as observations, not things we measured

Figure 5. Same issue of unclear y-axis as figure 3. Also, is “SVLxR:G” the same thing as “SVL*R:G”? If so, why use different characters for the interaction? If not, need a lot more clarification on what is being measured. This figure legend also needs details on the Collar Yes vs. No key. No explanation for this key is provided.

Statistical method has been changed after the advice of the 1st reviewer. So this figure has been removed.

Line 245: Recommend changing “classed” to “classified”.

Changed

Lines 252-254: I think these results are really interesting (i.e., that yellow coloration shows up prior to sexual maturation) and I would appreciate seeing more data to support this conclusion. It would appear that you probably have data on adult size and size of juveniles through development. Perhaps some sort of graph showing SVL on the y-axis over time on the x-axis, and a line or range showing the SVL/time when yellow coloration first shows up?

Unfortunately, we only have data taken every 4 weeks until they were approx. 16 weeks old. We did not have the time or resources to follow them further. It would be a very interesting study to complete in the future

Figure 6. Could you add the details about age and SVL to the figure itself? Perhaps as small inset details on the bottom of each image? This would make it a lot easier on readers instead of having to look back and forth between the figure and the figure legend for this information.

Figure amended and text changed

Table 2. I don’t think this table is all that informative given the goal of the experiment. Are we really that interest in the overall mean? Why not show graphs of the relationships of interest? i.e., the relationships between R:G ratio and measures of jump performance?

Graph showing relationship between R:G ratio and maximum distance travelled added. (Line 311)

Lines 300-309: I’m confused why this background information is presented in the results. Shouldn’t it be in the Introduction?

This was also pointed out by the other reviewer and has been removed as we decided it was unnecessary.

Lines 317-320: Could you add photos of the female throat of M. olmonae too? Might be a nice addition.

Unfortunately, the student that completed this is no longer in contact (has moved back to China)

Lines 324-30: I really think this Table S3 should be in the main text and not the supplementary. Otherwise, why even devote a results subsection to it? And I also think more summary details should be included in this results sub-section. For example, for how m any species has female yellow throat coloration been observed? Or male color change?

The table has been added as Table 3 and some summary details added

Line 340: “focused”

Technically UK English has the double ‘s’. However, this was changed to avoid confusion.

Lines 340-346: Please also provide a clear statement that your results supported the hypothesis that female throat coloration is related to female quality (i.e., yellower females jumped farther).

Added and clarified

Lines 332-346: A suggestion: I think this discussion could be a lot stronger if you provide clearer statements of what your study accomplished and what the key results were right in the beginning, without making the reader search for it later on.

Agreed, we have added a paragraph at the start.

Line 365: La Marca reference?

Corrected. It was an issue with a reference manager.

Line 390. Seems odd to suggest Zahavi as an afterthought with no real explanation. If this is a reasonable hypothesis, please provide a more clear explanation to how turning black when calling could be a “handicap”.

Removed as this may have been somewhat of a jump

Line 398. Please include the full genus name before the first Mannyphryne species listed, and “M.” before each subsequent species of the same genus.

Changed

Lines 401-407: I find this explanation based on phylogeny unclear. Are you saying that based on the presence of this trait among clades that is likely ancestral to the genus?

Clarified

Lines 412-414: Long sentence. I recommend moving reed frogs to the beginning so it doesn’t take so long to get to the subject.

Changed

Line 429: Change to “no one” 

Changed

Lines 465-467: Is this true? Allobates femoralis have been studied in quite some detail and males typically have several clutches in their territories and may take 1-2 days to transport tadpoles (round trip), if I recall correctly.

From what we have found and observed in our species, there is no evidence that this occurs. When we have located clutches in the wild (very rarely) they are only single clutches with up to 13 eggs. This may be a difference in that A. femoralis males are territorial and chase off intruders?

Lines 451-475: I was a bit surprised to hear the only concluding hypothesis is one about mutual mate choice (i.e., suggesting males may choose females based on yellow throat coloration). Didn’t Wells observe contests between females that seemed to be resolved with these yellow throat signals? Given these natural history observations, isn’t intrasexual communication of dominance associated with territory defense also a reasonable hypothesis for the function of these signals?

Agreed. I think over multiple revisions, our initial conclusions became amalgamated and then became clouded. We have highlighted the aspect of female to female intraspecific competition as part of the hypothesis for colouration 

Table S1. Please specify that the data presented are sample sizes (assuming that’s what they are).

Clarified

---

## [Decision Letter · Decision Letter 1]

31 Mar 2020

PONE-D-19-21985R1

Sexual dichromatism in the neotropical genus Mannophryne (Anura: Aromobatidae)

PLOS ONE

Dear Mr Greener,

Thank you for submitting your manuscript to PLOS ONE. After careful consideration, we feel that it has merit but does not fully meet PLOS ONE’s publication criteria as it currently stands. Therefore, we invite you to submit a revised version of the manuscript that addresses the points raised during the review process.

One of the original reviewers has reviewed the revised manuscript and found it to be improved. There are still some concerns regarding the figure and statistics that need to be addressed.

We would appreciate receiving your revised manuscript by May 15 2020 11:59PM. To enhance the reproducibility of your results, we recommend that if applicable you deposit your laboratory protocols in protocols.io, where a protocol can be assigned its own identifier (DOI) such that it can be cited independently in the future. For instructions see: http://journals.plos.org/plosone/s/submission-guidelines#loc-laboratory-protocols

We look forward to receiving your revised manuscript.

Kind regards,

Cheryl S. Rosenfeld, DVM, PhD

Academic Editor

PLOS ONE

Reviewers' comments:

Reviewer's Responses to Questions

**Comments to the Author**

1. If the authors have adequately addressed your comments raised in a previous round of review and you feel that this manuscript is now acceptable for publication, you may indicate that here to bypass the “Comments to the Author” section, enter your conflict of interest statement in the “Confidential to Editor” section, and submit your "Accept" recommendation.

Reviewer #2: (No Response)

2. Is the manuscript technically sound, and do the data support the conclusions?

Reviewer #2: Yes

3. Has the statistical analysis been performed appropriately and rigorously? 

Reviewer #2: Yes

4. Have the authors made all data underlying the findings in their manuscript fully available?

Reviewer #2: (No Response)

5. Is the manuscript presented in an intelligible fashion and written in standard English?

Reviewer #2: Yes

6. Review Comments to the Author

Reviewer #2: Overall, I’m happy with the edits that the authors have made to the manuscript to address my concerns, I just have three minor comments.

Lines 54-61: The edits to this section actually don’t include the function I suggested and appears to misrepresent the conclusions of the cited work. The conclusion of the Sztatecsny et al. (2012) study, along with several others that investigate explosive breeders, was that color change may function in sex recognition, not necessarily female choice – e.g., in the case of Rana arvalis, that turning blue allows males to be recognized as males by other males, so that they aren’t recipients of mistaken amplexus. This is different than color change having a function in female choice. The Sztatecsny et al. (2012) actually examined the potential for blue color to be attractive to females and found no evidence that females prefer blue males. This is an important distinction that is not made in the text.

Lines 128-129: The edits to the photograph protocol above are helpful, but in applying the color profile to other photographs, I’m assuming the profile was only applied to other photographs within a batch, and that each batch had it’s own color profile? By saying you applied the color profile to all photographs, this detail isn’t clear. Please clarify.

Lines 134-137: I agree that the linear mixed models are the better way to go, but please also state what tests were used to evaluate statistical significance, since these vary when implemented in R (e.g., wald chi-square tests, likelihood ratio tests? etc.)

7. PLOS authors have the option to publish the peer review history of their article (what does this mean?). If published, this will include your full peer review and any attached files.

Reviewer #2: No

---

## [Author Response · Author response to Decision Letter 1]

15 May 2020

6. Review Comments to the Author

Reviewer #2: Overall, I’m happy with the edits that the authors have made to the manuscript to address my concerns, I just have three minor comments.

Lines 54-61: The edits to this section actually don’t include the function I suggested and appears to misrepresent the conclusions of the cited work. The conclusion of the Sztatecsny et al. (2012) study, along with several others that investigate explosive breeders, was that color change may function in sex recognition, not necessarily female choice – e.g., in the case of Rana arvalis, that turning blue allows males to be recognized as males by other males, so that they aren’t recipients of mistaken amplexus. This is different than color change having a function in female choice. The Sztatecsny et al. (2012) actually examined the potential for blue color to be attractive to females and found no evidence that females prefer blue males. This is an important distinction that is not made in the text.

This has been addressed in the section and adapted to properly reflect on the results of the cited study.

Lines 128-129: The edits to the photograph protocol above are helpful, but in applying the color profile to other photographs, I’m assuming the profile was only applied to other photographs within a batch, and that each batch had it’s own color profile? By saying you applied the color profile to all photographs, this detail isn’t clear. Please clarify.

This has been clarified

Lines 134-137: I agree that the linear mixed models are the better way to go, but please also state what tests were used to evaluate statistical significance, since these vary when implemented in R (e.g., wald chi-square tests, likelihood ratio tests? etc.)

The test used to evaluate has been added to the section

---

## [Decision Letter · Decision Letter 2]

1 Jun 2020

Sexual dichromatism in the neotropical genus Mannophryne (Anura: Aromobatidae)

PONE-D-19-21985R2

Dear Dr. Greener,

We are pleased to inform you that your manuscript has been judged scientifically suitable for publication and will be formally accepted for publication once it complies with all outstanding technical requirements.

With kind regards,

Cheryl S. Rosenfeld, DVM, PhD

Section Editor

PLOS ONE

Additional Editor Comments (optional):

Reviewers' comments:

Reviewer's Responses to Questions

**Comments to the Author**

1. If the authors have adequately addressed your comments raised in a previous round of review and you feel that this manuscript is now acceptable for publication, you may indicate that here to bypass the “Comments to the Author” section, enter your conflict of interest statement in the “Confidential to Editor” section, and submit your "Accept" recommendation.

Reviewer #2: All comments have been addressed

2. Is the manuscript technically sound, and do the data support the conclusions?

Reviewer #2: (No Response)

3. Has the statistical analysis been performed appropriately and rigorously? 

Reviewer #2: (No Response)

4. Have the authors made all data underlying the findings in their manuscript fully available?

Reviewer #2: (No Response)

5. Is the manuscript presented in an intelligible fashion and written in standard English?

Reviewer #2: (No Response)

6. Review Comments to the Author

Reviewer #2: (No Response)

7. PLOS authors have the option to publish the peer review history of their article (what does this mean?). If published, this will include your full peer review and any attached files.

Reviewer #2: No

---

## [Editor Report · Acceptance letter]

16 Jun 2020

PONE-D-19-21985R2 

Sexual dichromatism in the neotropical genus Mannophryne (Anura: Aromobatidae) 

Dear Dr. Greener:

I'm pleased to inform you that your manuscript has been deemed suitable for publication in PLOS ONE. Congratulations! Your manuscript is now with our production department. 

Kind regards, 

on behalf of

Dr. Cheryl S. Rosenfeld 

Section Editor

PLOS ONE